# Temperate infection in a virus–host system previously known for virulent dynamics

Ben Knowles [1✉], Juan A. Bonachela [2], Michael J. Behrenfeld[3], Karen G. Bondoc [1], B. B. Cael[4], Craig A. Carlson[5], Nick Cieslik[1], Ben Diaz[1], Heidi L. Fuchs [1], Jason R. Graff[3], Juris A. Grasis [6], Kimberly H. Halsey [7], Liti Haramaty[1], Christopher T. Johns[1], Frank Natale[1], Jozef I. Nissimov[8], Brittany Schieler[1], Kimberlee Thamatrakoln [1], T. Frede Thingstad[9], Selina Våge[9], Cliff Watkins[1], Toby K. Westberry [3] & Kay D. Bidle [1✉]

The blooming cosmopolitan coccolithophore *Emiliania huxleyi* and its viruses (EhVs) are a model for density-dependent virulent dynamics. EhVs commonly exhibit rapid viral reproduction and drive host death in high-density laboratory cultures and mesocosms that simulate blooms. Here we show that this system exhibits physiology-dependent temperate dynamics at environmentally relevant *E. huxleyi* host densities rather than virulent dynamics, with viruses switching from a long-term non-lethal temperate phase in healthy hosts to a lethal lytic stage as host cells become physiologically stressed. Using this system as a model for temperate infection dynamics, we present a template to diagnose temperate infection in other virus–host systems by integrating experimental, theoretical, and environmental approaches. Finding temperate dynamics in such an established virulent host–virus model system indicates that temperateness may be more pervasive than previously considered, and that the role of viruses in bloom formation and decline may be governed by host physiology rather than by host–virus densities.

[1] Department of Marine and Coastal Science, Rutgers University, New Brunswick, NJ 08901, USA. [2] Department of Ecology, Evolution, and Natural Resources, Rutgers University, New Brunswick, NJ 08901, USA. [3] Department of Botany and Plant Pathology, Oregon State University, Corvallis, OR 97331, USA. [4] National Oceanography Centre, SO14 3ZH Southampton, UK. [5] Department of Ecology, Evolution, and Marine Biology, University of California Santa Barbara, Santa Barbara, CA 93106, USA. [6] School of Natural Sciences, University of California Merced, Merced, CA 95343, USA. [7] Department of Microbiology, Oregon State University, Corvallis, OR 97331, USA. [8] Department of Biology and the Waterloo Centre for Microbial Research, University of Waterloo, Waterloo, ON N2L 3G1, Canada. [9] Marine Microbiology Research Group, Department of Biological Sciences, University of Bergen, 5020 Bergen, Norway. ✉email: benjaminwilliamknowles@gmail.com; bidle@marine.rutgers.edu

Viruses routinely terminate phytoplankton blooms. This process is thought to be the outcome of density-dependent dynamics, where rising phytoplankton cell densities drive increased virus–host encounters and therefore infection. If these viruses are from virulent, purely lytic lineages, then they will rapidly replicate and kill their hosts upon infection regardless of host physiological state or environmental conditions. This makes virus–host encounters, infection, viral reproduction and host death equivalent in virulent systems, and allows virus–host dynamics and viral infection to be modeled theoretically as a direct outcome of readily quantifiable host and viral abundances: when host and viral densities rise, so does infection, lysis, and rates of lysis-mediated biogeochemical cycling[1–5]. As viral-mediated processes like the turnover and redistribution of energy and matter from lysed cells are driven by viral lysis[6,7], these models focus on how and under what conditions viruses kill their hosts. Altogether, this has led to a virulent-centric view of how viruses drive host diversification, global biogeochemistry, and termination of blooms, with host lysis and associated viral production commonly viewed as evidence of virulent infection.

However, viruses exhibit a spectrum of infection strategies from virulent to temperate. Like virulent viruses, temperate viral lineages are subject to density-dependent encounter-driven infection when they are extracellular, and can immediately initiate viral replication and host lysis upon successful infection. In addition to extracellular transmission, temperate viruses can become dormant after infection and be passed on intracellularly with replicating host cells. This effectively decouples infection from host death, minimizing deleterious environmental exposure[8], and allows viruses to time lysis for optimal reproduction and persistence[9]. Temperate infection outcomes are commonly dependent on host physiology rather than densities, with the viruses initiating replication and killing of their hosts ("induction") after sensing either host stress[10–12], extracellular conditions conducive to subsequent lytic infection[13], or loss of host ability to actively suppress resident viruses. Induction may occur at any host density. Although understudied in eukaryotic phytoplankton, extrapolation from temperate bacterial virus (bacteriophage) studies[8,14–20] suggests that temperate infection is more frequent than virulent infection in nature and likely drives biogeochemical cycling, host and viral diversification, and bloom dynamics in ways that are markedly divergent from those expected under virulent infection. Consequently, the mechanisms of temperate infection and induction stand to fundamentally revise our understanding of how viruses impact phytoplankton blooms, ecosystem dynamics, and host–virus evolution.

To our knowledge, all eukaryotic phytoplankton-virus systems examined to date have been characterized as virulent, thereby framing phytoplankton bloom dynamics within this virulent perspective (although lysogeny has been shown in cyanobacterial phytoplankton[21,22] and considered in free-living[23] and host-associated[24] eukaryotic algae). The cosmopolitan coccolithophore *Emiliania huxleyi* and its *Coccolithoviruses* (EhVs) are one of the central models of virulent dynamics. This virus–host system has provided critical insight into the subcellular molecular controls, global biogeochemical impacts, and ecosystem outcomes of virulent infection in terminating phytoplankton blooms[25–35]. Like other phytoplankton-virus systems, virulence in the *E. huxleyi*-EhV system has been supported by laboratory- and environmental-based studies showing lysis of hosts at high host densities or bloom climax that are matched by predictions from virulent-only theoretical models. Taken together, the *E. huxleyi*-EhV system, and other eukaryotic and prokaryotic virus–host systems, are characterized by multiple lines of evidence and inference as virulent and governed by density-dependent dynamics.

Recent work has shown the dominance of more-virulent phytoplankton viruses in the laboratory and less-virulent viruses in the environment[28]. Building on this observation, we sought to determine if the "rules of infection" in the *E. huxleyi*-EhV system were conserved across nutrient-enriched laboratory and mesocosm conditions with host densities of ~$10^5$–$10^6$ cells per milliliter and environmental conditions with densities of ≤$10^3$ cells per milliliter. In laboratory experiments, the most virulent viruses in our culture collection[28,36] showed virulent phenotypes at high initial host densities and temperate, physiology-dependent dynamics at natural initial densities where induction was triggered by cellular stress, consistent with other microbiological systems[8,10]. Theoretical modeling and statistical analysis resolved this seeming paradox by showing that high host densities trigger induction, making temperate viruses behave like virulent viruses at these high densities as they initiate lysis rapidly after establishing infection. Guided by these insights, we revisited data sets from nutrient-enriched, high-density environmental mesocosms established as models of virulent infection[31] and found that natural *E. huxleyi* and EhV populations also exhibit temperate infection dynamics. As a result, even apparently virulent infection dynamics are equally likely to be temperate in this model virlent virus–host system. This provides a mandate to revisit whether other seemingly virulent virus–host systems possess temperate-ness using our analytical template that integrates laboratory, theoretical, and environmental analyses. Taken together, we present an empirically derived physiology-dependent model of phytoplankton bloom and decline governed by "bottom-up" limitation triggering "top-down" control. In this model, temperate viruses transition from pervasive, long-term, non-lethal temperate infection of healthy hosts (virus–host "*Détente*") as blooms form to killing off ailing host populations once they become physiologically stressed as blooms peak (viral-mediated "*Coup de Grâce*").

## Results and discussion

**Infection at experimental and natural *E. huxleyi* densities**. The *E. huxleyi*-EhV system is a fundamental model of virulent infection dynamics. Infection in high-density lab cultures and nutrient-amended, eutrophic mesocosms ($10^5$–$10^6$ cells per milliliter; "*crowded*" densities) result in host death and bloom termination within days[25,31,37–39]. In stark contrast, natural *E. huxleyi* bloom densities across the global ocean are almost exclusively $10^3$ cells per milliliter or less (hereafter referred to as "*sparse*" densities; Fig. 1a, left panel inset; Supplementary Fig. 1), where infection and viral-mediated lysis of hosts has been observed[34,40].

**Temperate infection in the *E. huxleyi* system**. To determine whether virulent "*rules of infection*" are conserved across disparate experimental and natural host densities, we conducted experiments under constant environmental conditions for *E. huxleyi* cell densities ranging from ~$10^1$ to ~$10^6$ cells per milliliter (Supplementary Table 1). We coincubated *E. huxleyi* strain CCMP374 cells that are highly sensitive to infection[27,35] with the highly virulent EhV207 virus[28,36] under light[41] and nutrient (*f*/2 media)[42] conditions known to promote host growth and virulent infection at a multiplicty of infection (MOI) of 10:1 viruses:host. Given that viruses terminate *E. huxleyi* blooms at ~$10^3$ cells per milliliter in nature[34,40], we expected that infection would limit host densities to similar levels regardless of initial host densities or conditions under the assumption that density-dependent virulent dynamics should be driven by densities alone (Supplementary Figs. 1 and 2; see Supplementary Fig. 3 for an overview of experimental inference). However, no viral-mediated death was

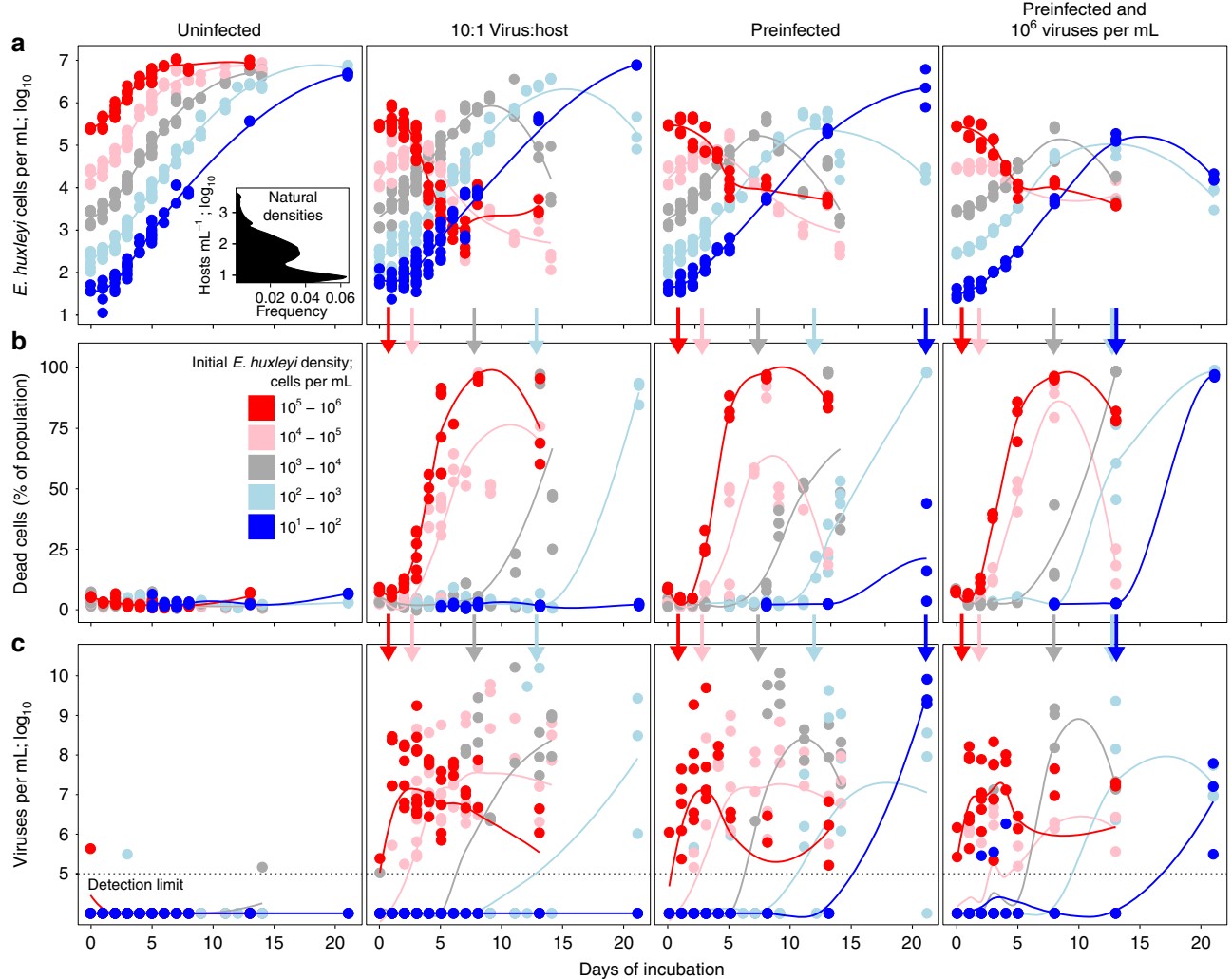

**Fig. 1 Cultures show dormant infection at natural densities and lysis at unnaturally high densities.** Time course of **a** host densities ($\log_{10}$-transformed; $n = 1339$ counts across seven independent experiments), **b** proportion of dead host cells ($n = 948$ counts across five independent experiments; SYTOX positive; data points with >$10^3$ cells per milliliter threshold) and **c** extracellular viral densities ($\log_{10}$-transformed; $n = 1188$ counts across four independent experiments) in populations of uninfected hosts, infected 10:1 virus:host multiplicity of infection (MOI) coincubations, and pre-infected hosts without or with extracellular viruses across a range of initial host densities ($10^1$–$10^2$ (blue), $10^2$–$10^3$ (light blue), $10^3$–$10^4$ (gray), $10^4$–$10^5$ (pink), and $10^5$–$10^6$ (red) host cells per milliliter points and lines of best fit, respectively). Inset histogram in **a** shows satellite-derived pixel-wise maximum *E. huxleyi* densities in the global ocean >100 m depth between 2003 and 2017; see Supplementary Fig. 1). Vertical arrows show average onset of host lysis in each density treatment. Data in **a** and **b** were generated by flow cytometry; **c** by qPCR (see Supplementary Fig. 13d for standard curves and thresholds). Independent experiments are summarized in Supplementary Table 1. Source data are provided as a Source Data file.

observed at natural host densities of ≤$10^3$ cells per milliliter. Rather, cultures only showed lytic declines once they had reached ~$10^5$ cells per milliliter, regardless of initial density treatment and across multiple independent experiments and experimental designs (Fig. 1a–c second panel; Supplementary Table 1). Crowded cultures with unnaturally high initial densities ≥$10^4$ cells per milliliter exhibited rapid host lysis, an increased frequency of dead cells, and elevated extracellular viral concentrations. In contrast, sparse cultures with initial densities ≤$10^3$ cells per milliliter exhibited substantial delays in viral lysis of hosts and only showed significant lysis after host densities reached ~$10^5$ cells per milliliter (Fig. 1a–c, second panel). These results are inconsistent with virulent dynamics given that EhVs can infect and kill at natural *E. huxleyi* densities[34,40].

Results from our sparse culture conditions suggested that EhVs may engage in temperate behavior, where lysis does not immediately follow infection. To test this, we pre-infected hosts by co-incubating ~$10^6$ hosts and ~$10^7$ viruses per milliliter for

2 h, resulting in a ~99% infection estimated by modeling or 40 ± 5% mean ± SE infection measured empirically (Supplementary Fig. 4) and then removed extracellular viruses by centrifugation and repeated washing. This treatment rendered host–virus encounter rates inconsequential during the initial stages of the subsequent growth phase (Fig. 1a–c; third panel). To a subset of these pre-infected cultures, we added an additional $10^6$ viruses per milliliter (tenfold higher than natural EhV densities[40]) to ensure infection at all initial host densities. This treatment may even promote dormant infection in temperate systems via a high MOI and multiple infections per cell[11,43] (Fig. 1a–c; right panels). Given the previously observed virulence of EhV207[28,36], we expected that nearly all *E. huxleyi* cells would die in a few days at all culture densities (Supplementary Figs. 2 and 3). Instead, we observed a remarkable lack of death and viral production in sparse cultures despite successful infection and host declines in crowded cultures (Fig. 1; third and fourth columns).

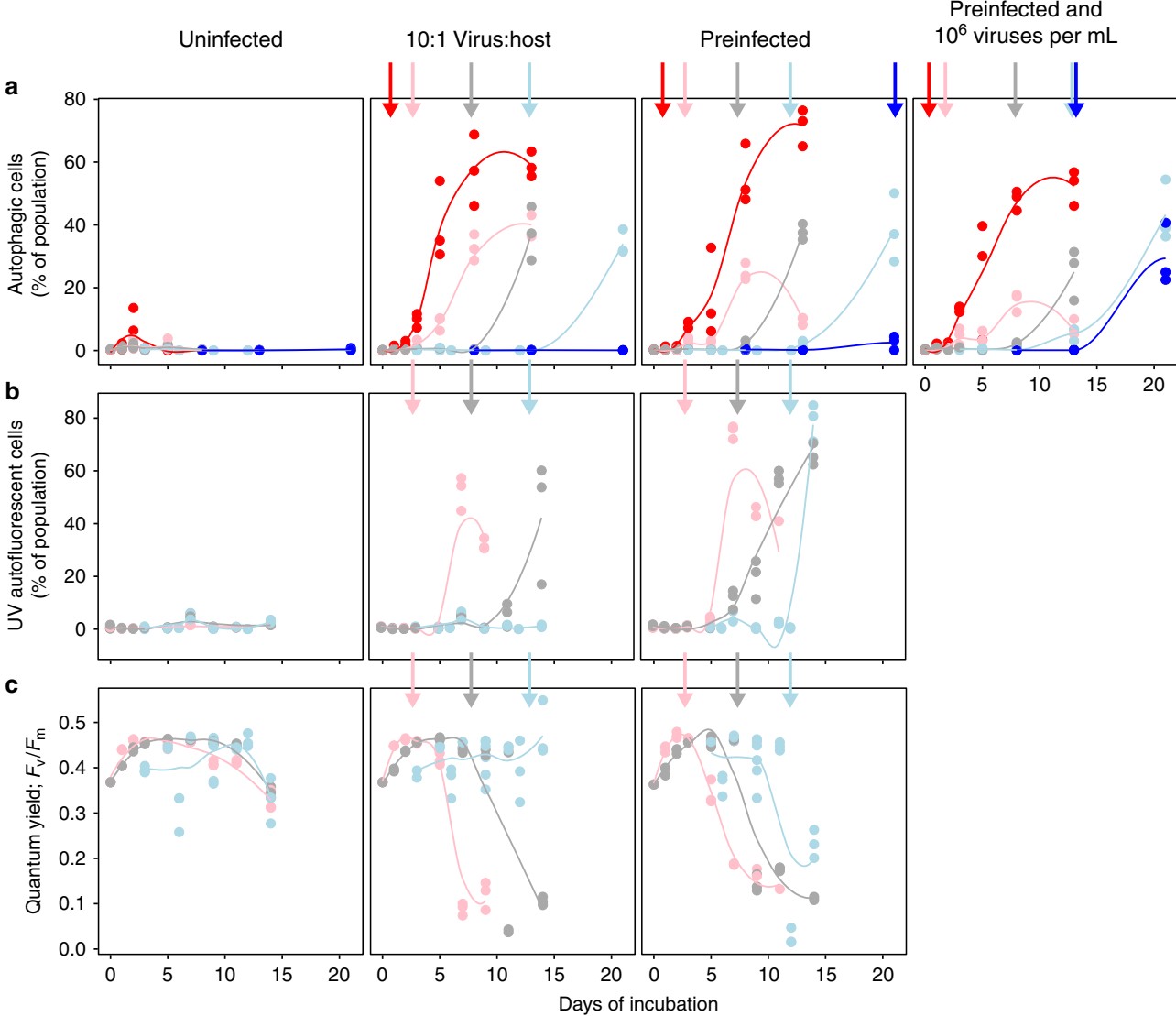

**Fig. 2 Infection remains dormant until the onset of host physiological stress.** Host physiological stress as the percent of cells **a** undergoing autophagy (Lysotracker stain positive; $n = 534$ counts across two independent experiments), **b** exhibiting elevated ultra-violet autofluorescence from metabolic dysfunction ($n = 295$ counts across two independent experiments), or **c** showing declining population photochemical quantum yield of PSII ($F_v/F_m$; $n = 284$ counts across two independent experiments) across uninfected, infected 10:1 virus:host multiplicity of infection (MOI) co-incubations, and pre-infected hosts without or with extracellular viruses added treatments. All analyses had a $10^3$ cells per milliliter detection threshold (see Supplementary Fig. 13b, c). Initial host density treatments span $10^1$–$10^2$ (blue), $10^2$–$10^3$ (light blue), $10^3$–$10^4$ (gray), $10^4$–$10^5$ (pink), and $10^5$–$10^6$ (red) host cells per milliliter points and lines of best fit, respectively. Vertical arrows show average onset of host lysis in each density treatment. Independent experiments are summarized in Supplementary Table 1. Source data are provided as a Source Data file.

There was also a lack of cellular stress in infected cultures prior to reaching crowded densities and the onset of lysis (Fig. 2). Rapid increases in the fraction of autophagy-positive cells upon induction (Fig. 2a) were consistent with both the onset of host stress and activated viral replication[29]. We serendipitously discovered that induction also resulted in increased cellular ultra-violet induced autofluorescence consistent with NADPH accumulation in infected cells, as viruses divert cellular resources towards the oxidative pentose phosphate pathway, nucleotide biosynthesis, and viral genome replication[41,44,45]. This marker of active intracellular replication was not observed prior to induction (Fig. 2b). Finally, photochemical quantum yield ($F_v/F_m$), a sensitive indicator of cellular and photochemical stress, showed that infected hosts remained healthy at sparse culture densities and then exhibited signs of physiological stress at crowded densities (Fig. 2c). These multiple, independent lines of

evidence demonstrate the presence of temperate behavior in the *E. huxleyi*-EhV system which is, to our knowledge, the first observation of temperate infection in eukaryotic phytoplankton. Indeed, lysis was only observed under highly constrained circumstances. Non-lethal infection and relaxed lysis ("*Détente*") was the norm in virus–host interactions, especially at environmental densities.

**Modeling of virulent and temperate infection**. We compared our empirical host dynamics with those predicted from two distinct theoretical models. The first was a virulence model and the second was a phenomenological temperate model in which lysis is triggered at the time-point when the experimental data show the onset of host photochemical stress (a declining $F_v/F_m$; Figs. 2c and 3 and Supplementary Fig. 2). The empirical results were qualitatively and quantitatively reproduced by the phenomological

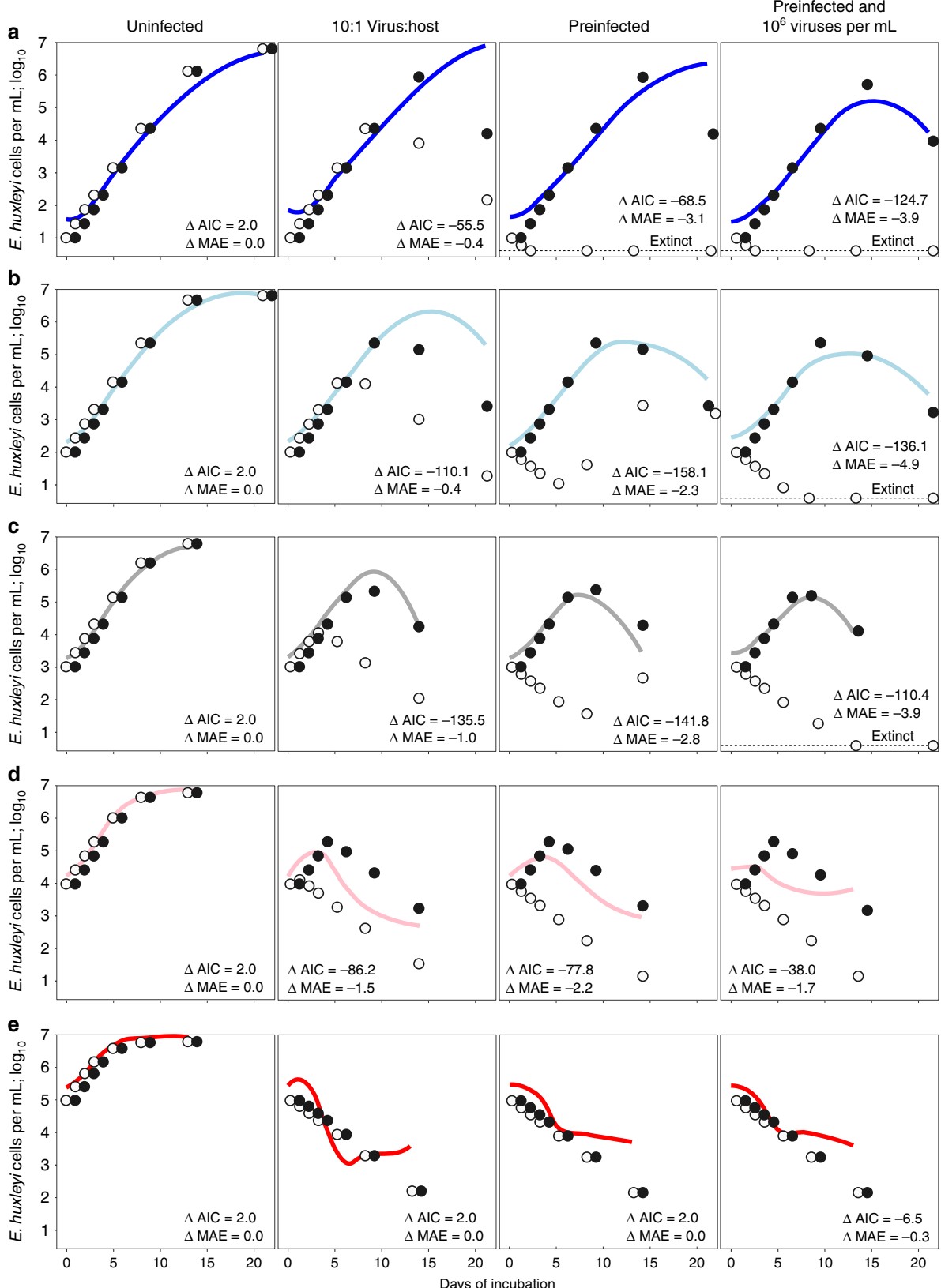

temperate model across all treatments and initial densities, but not by the virulence model, further providing further evidence of temperate infection (Fig. 3 and Supplementary Fig. 2; negative differences in akaike information criterion and mean absolute error values between models—ΔAIC and ΔMAE, respectively—

indicate better fits to empirical data for the temperate model than for the virulent model)[46,47]. Although the phenomenological temperate model matched the empirical data under most conditions, the low ΔAIC values were also driven by how poorly the virulence model fit the data (Fig. 3).

**Fig. 3 Culture dynamics reflect temperate rather than virulent infection.** Empirical data for *E. huxleyi* cell densities (lines of best fit from Fig. 1a) and theoretical predictions from virulence (open circles) and phenomenological temperate (black circles) models across initial host densities from **a** $10^1$-$10^2$ (blue), **b** $10^2$-$10^3$ (light blue), **c** $10^3$-$10^4$ (gray), **d** $10^4$-$10^5$ (pink), and **e** $10^5$-$10^6$ (red) host cells per milliliter in uninfected, 10:1 virus:host multiplicity of infection (MOI) co-incubations, and pre-infected hosts without or with extracellular virus treatments are shown. Empirical lines of best fit from Fig. 1a are colored by initial host density as presented in Fig. 1. Model-predicted extinct host populations where host densities fell ≤1 cell per milliliter are shown on the dashed horizontal lines. Difference in akaike information criterion ($\Delta$AIC; $AIC_{temperate}$ – $AIC_{virulent}$) and mean absolute error ($\Delta$MAE; $MAE_{temperate}$ – $MAE_{virulent}$; $log_{10}$) between models are shown, where negative values indicate that temperate models fit empirical data (Fig. 1a) better than virulent models despite being penalized for having extra parameters. Source data are provided as a Source Data file.

Virulence and temperate theoretical model predictions are strikingly divergent at low initial host densities (Fig. 3a–d). However, they are indistinguishable at the host densities of ≥$10^5$ cells per milliliter that are commonly used in laboratory and mesocosm experiments (Fig. 3e)[25,27,31,35,38] because temperate viruses rapidly initiate lysis when hosts become stressed at these high densities. Under these conditions, the viruses act as if they are virulent despite having the capacity to be temperate. As a result, temperate and virulent dynamics are equally likely to describe lysis when crowded densities and stress are conflated, as shown by the similar AIC values between temperate and virulent models in Fig. 3e. Previous characterizations of EhV207 as virulent were made in experiments under such conflated conditions. This explains why we were only able to discern temperateness in this system by conducting experiments across a range of host densities and physiological states.

To better understand the potential of host cellular stress to trigger induction, we implemented a modified version of the phenomenological temperate model in which we replaced the data-imposed induction timing used above with a self-regulated induction time that emerges from host growth dynamics (as in Fig. 2; Supplementary Fig. 5; see Supplementary Notes 1 and 2 for details and Supplementary Table 2 for parameters). Rather than being phenomenologically based on the timing of the *E. huxleyi*-EhV system, this model is broadly generalizable because it is informed by the onset of the autophagy stress response (e.g., Fig. 2a) that is involved in viral infection across eukaryotic host–virus systems[48], including *E. huxleyi*-EhVs[29]. The self-regulated induction model showed fits to the empirical data comparable to those of the phenomenological temperate model, and far closer than the virulence model despite being penalized for having more parameters (shown by the negative AIC values in Supplementary Fig. 5). This finding reinforces the importance of intracellular controls of temperate infection. The model thus complements our experimental approach (e.g., Fig. 1) by providing a template to diagnose temperate infections in other ecologically relevant, phytoplankton-virus model systems like diatoms, chlorophytes, or cyanobacteria in which virulence has also been established using elevated host densities.

**Physiological stress triggers induction of temperate viruses**. We built upon the agreement between our temperate theoretical models and empirical data by further investigating the role of stress as a driver of induction. We conducted experiments where hosts were inoculated either into low-nutrient seawater or in standard nutrient-rich *f*/2 media. The seawater incubations showed up to 1000-fold lower stationary phase densities ("carrying capacity"; Supplementary Fig. 6) than corresponding *f*/2 incubations (Fig. 4a, compare Supplementary Fig. 7 and Fig. 1a) and similar 1000-fold differences in the host density at which viruses initiated lytic induction ("lytic density") in infected cultures (Fig. 4a and Supplementary Fig. 7). Lysis was detected in infected cultures at time points when parallel uninfected control cultures showed slowing growth and onset of stress, regardless of host density (Fig. 4a and Supplementary Fig. 8). These observations

suggest that induction is dependent on host physiology rather than cell density, an idea supported by our temperate models of seawater incubations (Supplementary Fig. 9). Consistent with this concept, lytic density scaled with carrying capacity (Fig. 4b), presenting a means to distinguish physiologically sensitive temperate dynamics from physiologically insensitive virulent dynamics via experimental manipulation of carrying capacities. The scaling of lytic density with carrying capacity also explains why, after a notable lack of viral-induced mortality at natural host densities in high carrying capacity *f*/2 media in all prior experiments (Figs. 1 and 2), we observed that EhV207 killed *E. huxleyi* at natural densities in stressful, low carrying capacity seawater incubations (Fig. 4b and Supplementary Figs. 7 and 8).

**Temperate infection in the environment**. Distinguishing temperate and virulent infection in the environment is challenging because carrying capacities, the timing of infection, etc., are often unknowable. Given this uncertainty and that virulence dynamics and stress-driven induction of temperate infections are open to conflation, it is possible that any virulent infections or instances of viral-mediated lysis identified to date may actually be temperate. This makes detecting whether and when infection has taken place centrally important. In addition to sequence- and tracer-based techniques and chemically induced induction[11,15,49–53], the positive relationship between carrying capacity and lytic density (Fig. 4) presents a experimental template to extricate these infection strategies by identifying temperate dynamics. Using our logic from above (Fig. 1), if alleviation of bottom-up limitation by experimental provision of elevated nutrients, light, vitamins, etc. allows hosts to attain densities higher than bloom densities or densities at which viruses are known to infect and kill, which is ~$10^3$ cells per milliliter in *E. huxleyi*[34,40], then top-down virulent control is not prevalent in that system (Fig. 4). Further, virulent and temperate infection strategies differ through the timing of killing relative to the timing of infection: rapid host mortlity after virulent infection contrasted with rapid to non-existent host-mortality after temperate infection. Virulent and temperate infections can, therefore, be diagnosed by contrasting minimum densities at which infection is known to occur (~$10^3$ or less in the *E. huxleyi* system;[34,40]) with the density at which populations were "terminated" under nutrient replete conditions (Fig. 4).

The coincidence of rapid host death and viral production during bloom collapse is generally accepted as evidence of virulent control of host populations. However, it can also be the outcome of temperate viruses inducing and lysing stressed hosts (Fig. 3). We re-examined data sets from a nutrient-amended, coastal *E. huxleyi*-EhV mesocosm experiment (Fig. 5), which had been previously characterized to display virulent dynamics[31] using logic that underpins the virulent perspective in viral ecology[25,26,31,39,54,55]. These mesocosms had appeared to show periods where (*i*) host scarcity precluded infection during "lag" phase; (*ii*) hosts became dense enough to suffer viral encounter and infection, and (*iii*) hosts died with associated production of viruses and virus-associated lipids[31].

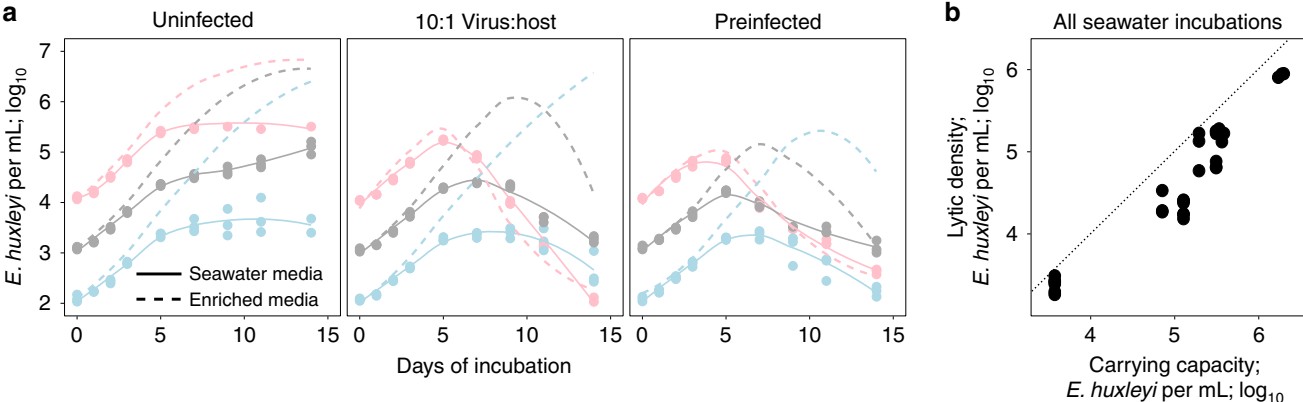

**Fig. 4 Physiological stress triggers induction. a** *E. huxleyi* growth curves in a representative experiment (Supplementary Table 1) in paired unenriched seawater (solid lines of best fit and data points; $n = 3$ independent counts in each treatment per time-point) versus $f/2$ enriched media (dashed lines of best fit from Fig. 1a) incubations in uninfected, 10:1 virus:host multiplicity of infection (MOI) co-incubations, and pre-infected hosts treatments. Similar experiments were conducted four times (Supplementary Table 1). **b** Lytic density varies as a positive function of carrying capacity across all seawater incubations ($n = 30$ counts across four independent experiments; Supplementary Table 1). Lytic density is the host density at which viral-induced mortality was observed in each infected culture, and carrying capacity is the density at which uninfected controls entered stationary phase for a given experiment and initial density (Supplementary Fig. 6). Data points and lines of best fit in **a** are colored by initial host density (lines of best fit for $10^2$–$10^3$ (light blue), $10^3$–$10^4$ (gray), and $10^4$–$10^5$ (pink) initial densities, respectively). Dashed diagonal 1:1 line is shown in **b**. Source data are provided as a Source Data file.

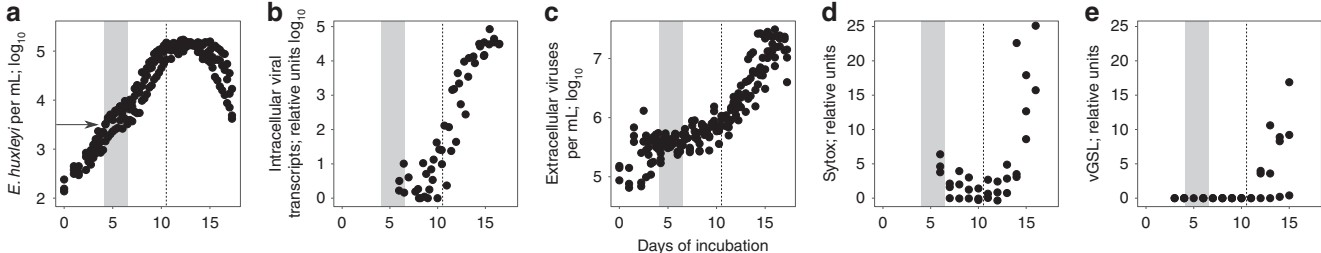

**Fig. 5 Temperate dynamics in natural *E. huxleyi* populations.** Dynamics in **a** *E. huxleyi* cell abundance (sampled days 0–18; $n = 191$ counts across three replicate mesocosms), **b** intracellular EhV transcription (relative units; Pagarete et al.[55]; units from original paper; $n = 50$ counts across three replicate mesocosms), **c** extracellular EhV densities (sampled days 0–18; $n = 185$ counts across three replicate mesocosms), **d** *E. huxleyi* cell death (relative fluorescence of SYTOX viability stain; sampled days 6–16; $n = 21$ independent counts across two replicate mesocosms), and **e** EhV-derived viral glycosphingolipid (vGSL; relative units; sampled days 3–16; $n = 24$ counts across two replicate mesocosms) in coastal Norwegian mesocosms. All data are in units from the original publications: **a** and **c**–**e** from ref. [31], and **b** from ref. [55]. Horizontal arrow in **a** represents the maximal environmental *E. huxleyi* densities where viral infection and lysis have been observed (~$10^3$ to $10^{3.5}$ *E. huxleyi* cells per milliliter;[34,40]; Supplementary Fig. 1). All panels show the window of time in which replicate mesocosms surpassed environmental densities where the onset of density-dependent infection and subsequent lysis was expected to occur (i.e., within the gray shaded boxes) and when bloom growth stalled (dashed veritcal lines) in contrast with when death, viral production, and host lipid remodeling were observed. Source data are provided as a Source Data file.

If this system showed density-dependent, physiology-insensitive virulent infection, we would expect to observe death at host densities of ~$10^3$ cells per milliliter[34,40]. In contrast, if infection was temperate and physiology-dependant, cells could grow beyond these densities. Indeed, log-transformed data revealed an absence of lag phase but instead exponential growth from the start of the experiment with rapidly proliferating hosts growing to $10^5$ cells per millilter, surpassing maximal environmental densities by ~100-fold, and growing for ~4 days after passing densities at which density-dependent lysis would be observed if the system were virulent (Fig. 5a and Supplementary Fig. 1). This exponential growth occurred even though early detection (day 6 of 18) and steadily increasing intracellular EhV transcripts were observed (Fig. 5b). Despite this evidence of early infection, lysis was delayed for ≳1 week before extracellular EhV concentrations (Fig. 5c), host cell death (Fig. 5d), and production of virus-specific lipids[31,55,56] increased (Fig. 5e). Further, the onset of lysis only occurred after the bloom had shown stationary phase dynamics for ~3 days, consistent with viruses lysing cells

that were stressed, senescing, and/or dying, again consistent with temperate infection. Temperate infection appears to be prevalent—albeit undiagnosed—in natural *E. huxleyi*-EhV communities.

**Propagation of infection.** EhVs lose infectiousness in ~3 days (Supplementary Fig. 10a). However, physically explicit theoretical estimates suggest that it takes newly produced EhVs >10 days after lysis to encounter a new *E. huxleyi* host under environmental host densities and turbulence regimes (Supplementary Fig. 10b, c). Extracellular transmission in nature is further hindered by adsorption to transparent exopolysaccharides[57,58] and free coccoliths[59], aggregation and vertical export of infected cells[34], and active anti-viral-propagation strategies[56]. Temperate propagation within hosts can evade these barriers and enable viral populations to persist and co-exist with their hosts[60], especially during periods of host scarcity between blooms. It also suggests that lysis may be a strategy of last resort because switching from intracellular to extracellular virus transmission exposes viral progeny to persistent challenges posed by the extracellular environment[8]. Notably, these

barriers to virulent transmission across temporal and spatial scales are likely universal. As a result, viruses of unicellular eukaryotic phytoplankton with natural cell densities similar to *E. huxleyi* may similarly require temperateness for propagation, despite being thought of as virulent predators based on previous culture studies conducted at host densities far exceeding those observed in nature.

Indeed, pervasive temperate infection may resolve the long-standing question of how viral infection can propagate across micro- to meso-scales (~100 μm to ~100 km, respectively)[61] within the week to month time frames of bloom formation and result in observed, near-coordinated host collapses within days of bloom climax[8,30,34]. By allowing delays between infection and host lysis, temperate infection is also consistent with facilitated particle aggregation and preferential sinking of early, pervasively infected cells from the upper mixed layer to deeper mesopelagic waters ("viral shuttle")[30,32,34,58,62], as opposed to the massive release of cellular dissolved organic matter within the upper mixed layer associated with virulent infection via the "viral shunt"[6]. Indeed, our work suggests that the viral shunt may itself be the outcome of cellular stress triggering lytic induction of temperate infections. Incorporating temperate dynamics into these "viral shunt"/"viral shuttle" pathways as a physiology-dependent ecosystem process can conceptually and quantifiably couple top-down and bottom-up processes in response to nutrient dynamics.

## Bloom formation and decline is driven by host physiology.
The discovery of temperate infection in the *E. huxleyi*-EhV system restructures our thinking of the nature of viral–host interactions, propagation, and persistence. Our evidence presents a new conceptual model of bloom formation and decline wherein bloom formation can commence under relaxed control of host populations by temperate viruses (virus–host "*Détente*") and terminate with declining host growth rates prompting lysis by those viruses (viruses deliver a "*Coup de Grâce*"; Supplementary Fig. 11) with resulting viral production and host death. This model emphasizes the importance of host physiology to viral infection dynamics, opening new avenues of research from molecular and genetic to global spatial scales and from ecological to evolutionary temporal scales. For example, although it appears that the *Coup de Grâce* model shares stress-related physiological induction with bacteriophage microbiological and environmental models[8,10,11,15,16], these dynamics play out across dissimilar respective host densities of ~$10^3$ and ≥$10^6$ phytoplankton and bacterial cells per milliliter, and different environments (open ocean compared to coral reefs, lung, and gut). Thus, although evidence suggests that virus–host interactions may be governed by universal physiological "rules", elucidating the molecular and physiological cues that trigger induction and their relationship to host densities is central to mechanistic, predictive ecosystem modeling efforts.

Refocusing from dynamics driven by host population densities, the Coup de Grâce model suggests that viral infection dynamics play out at the level of individual cells. In this view, each cell presents a unique physiological state and infection outcome, especially in heterogeneous environments. It further posits that in the absence of lysis of healthy cells, microbial cell densities may rise and be sustained longer under temperate infection dynamics (Fig. 3 and Supplementary Fig. 11). This could effectively localize energy and biomass within the particulate/microbial fraction, fundamentally bypassing the "viral shunt", whereby organic matter is attenuated and respired in surface waters[6]. There may also be currently unexamined differences in cellular products liberated in the viral shunt through host mortality by virulent lysis of proliferating cells compared to by induction in ailing populations.

Ultimately, the *Coup de Grâce* model provides a rationale to revise how viruses influence the flow of energy and matter through ecosystems, and how processes like viral infection and intracellular "decision-making" operate in nature to impact bloom formation and demise. Thinking across different temporal scales, temperateness enables virus–host persistence year-around and presents a more evolutionary stable host–virus landscape[60], the advantages and tradeoffs, which remain unknown. In heterogeneous populations, though, it is unknown what fraction of host populations induce during bloom termination, whether viruses produced by "last resort" induction are able to infect subsequent hosts or decay, and what fraction of hosts—harboring dormant viruses—go on to maintain populations that persist between blooms. Taken together, the biogeochemical, ecological, and evolutionary impacts of virulent versus temperate infection dynamics are open questions of global importance.

## Methods

**Host cultures**. All laboratory experiments were conducted with *Emiliania huxleyi* strain CCMP374 (https://ncma.bigelow.org/ccmp374) and EhV strain 207 (see below). CCMP374 is a naked strain of *E. huxleyi* isolated from the Gulf of Maine in 1990, and exhibits rapid growth, high-stationary phase densities (~$10^7$ cells per milliliter), and high sensitivity to viral infection[27,63]. Cultures were maintained at 5·$10^5$ to 1·$10^6$ cells per milliliter and grown at 18 °C, with a 14 h:10 h light:dark cycle with a light intensity of 125 μmol photons m$^{-2}$ s$^{-1}$. CCMP374 was grown in batch culture conditions with *f*/2 rich nutrients[42] added to 0.2 μm pore-size filtered (GE Healthcare USA, filter 6718-9582) autoclaved seawater in either polystyrene 50 mL flasks or 6-well plates or polypropylene 96-well plates (Greiner Bio-One, USA; items 690160, 657185, and 780270, respectively; Supplementary Table 1). Addition of *f*/2 nutrients increases macronutrient concentrations (e.g., NaNO$_3$ 882 μM; an ~88-fold enrichment over basal seawater with ~10 μM NaNO$_3$; other nutrients see similar enrichments). This provides ideal, replete conditions conducive to virulent dynamics in which to probe for the presence of virulent viral behavior.

**Virus cultures**. EhV207 has commonly been used to elucidate virulent dynamics, as it induces the rapid decline of host populations and concomitant production of high titers of viral progeny under culture conditions[28,36,64]. Together with CCMP374, EhV207 comprises a highly virulent host–virus system, strongly predisposing this work towards the execution of virulent activity. Viruses were cultured by adding them to exponentially growing cultures at ~5·$10^5$ to 1·$10^6$ cells per milliliter in *f*/2 media at a virus:host ratio of 10:1 MOI. Cultures visibly cleared after approximately three days and viruses were isolated from cellular debris using 0.45 μm pore-size filtration (EMD Millipore, USA; filters SLHV033RS or SVHV01015) and lysates stored in the dark at 4 °C until use within 1 week. This approach yielded viral titers in excess of $10^8$ viruses per milliliter. In experiments where a virus-negative control was required, a heat-killed lysate was produced by incubation at 90 °C for 10–20 min prior to 0.02 μm pore-size filtration (Anotop, Whatman, USA) and cooling to ~18 °C. All infections were conducted in the morning[41]. For all experiments, virus infectivity was monitored by running parallel cultures with initial host densities of $10^5$ cells per milliliter coincubated with a MOI of 10 (10:1 viruses:host). These visibly cleared in all cases, showing that our viruses were always infectious in these experiments. All flasks were shaken daily and plates mixed by pipetting to preclude settling and ensure equal exposure to infection. In summary, all experiments were conducted in a manner typically conducive to virulent infection and with viable viruses and sensitive hosts.

**Laboratory coincubation experiments**. *E. huxleyi*-EhV virulent infection dynamics were first were studied using coincubation of viruses and hosts at an initial ratio of 10:1 viruses:host (MOI = 10) in laboratory conditions. In experiments without preinfection treatments—Experiments II, III, and VII—viral lysates were added to high-density (~1·$10^6$ cells per milliliter; quantified using a Coulter Counter Multi-sizer 3, Beckman, USA) cultures to a final ratio of 10:1 virus:host (MOI = 10; viruses were quantified using an Influx Mariner flow cytometer; BD, USA). Cultures were then serially diluted down to experimental densities; all cells were from the same inoculum within each experiment (see Supplementary Table 1 for densities in each experiment and Supplementary Fig. 3 for experimental rationale). Uninfected controls substituted lysates with heat-killed, filtered lysate. In experiments with preincubation treatments (Experiments I, IV, V, and VI), cultures for 10:1 MOI coincubation treatment were drawn from the uninfected control after centrifugation and washing, so that uninfected controls, 10:1 MOI coincubation, and pre-infected treatments were all subjected to similar centrifugation and washing before lysate/heat-killed viral addition. The initial set of experiments (Experiments I, II, III, and VII) were conducted for approximately a week as we expected lysis to occur at all densities in that time (Supplementary Table 1). These experiments were subsequently repeated due to our initial interpretation that the lack of lysis in scarce densities (<$10^4$ cells per milliliter) was spurious and

susceptible to resolution with further experiments. Discovering instead that this lack of lysis is robust, we commenced longer experiments to demonstrate that the lack of death is not from declining viral infectiousness or abundances, as death initiated up to 3 weeks into incubations (Experiments IV, V, and VI; Supplementary Table 1).

**Laboratory pre-infected and virus addition experiments.** Having established that lack of lysis in low-density treatments in *f*/2 media was a robust phenotype in coincubations, we sought to determine if it arose from viruses either being unable to infect at these densities or from viruses choosing not to kill infected hosts at these densities. To distinguish these scenarios, hosts were pre-infected at high density before dilution (Experiments I, IV, V, and VI; Supplementary Table 1). High density (~1·10^6 cells per milliliter; quantified using a Coulter Counter Multisizer 3, Beckman, USA) host cultures were coincubated with a tenfold higher concentration of virus (~1·10^7 EhVs per milliliter, final density; quantified using an Influx Mariner flow cytometer; BD, USA) for 2 h in culture flasks (Greiner Bio-One, USA) under standard culture conditions (see above). Uninfected controls and coincubation treatments (see above; these treatments were pooled at this point) received a similar volume of heat-killed, filtered lysate. Cultures were pelleted at speed 7 (30 cm radius, swing bucket rotor; Fisher Scientific Centrific Model 225 centrifuge; ~5000 r.c.f.) for 10 min in 50 mL polypropylene tubes (Corning, USA, item 352070). This was done three times so that samples were "triple washed" of extracellular viruses. Pelleting did not affect cellular health as uninfected controls at the start of experiments ($t_0$) showed low levels of stress and vigorous growth after resuspension. Supernatants were discarded, and tubes inverted for ~3 min to remove as much supernatant and as many free viruses as possible. Pelleted cells were resuspended in fresh *f*/2 media, vortexed and transferred to new tubes between spins. After washing, cell densities were quantified (Coulter Counter Multi-sizer 3, Beckman, USA) and serially diluted to experimental densities using fresh media. After dilution down to experimental densities, viruses were also added to a preinfection treatment flasks (10^6 virus per milliliter; final density). All seawater incubations were preceded by pelleting cells growing in *f*/2 nutrient-rich media and resuspending them in unamended filtered autoclaved seawater, rapidly transitioning cells from *f*/2 rich media to seawater.

**Flow cytometry setup.** Laboratory-based experiments were conducted at the Rutgers University Microbial Flow Sort Facility (https://marine.rutgers.edu/microbial-flow-sort-lab/) using a BD Influx Mariner 209s flow cytometer equipped with 355, 488, and 640 nm excitation lasers. All samples were vortexed immediately prior to being run on the flow cytometer, and flow rates measured repeated throughout measuring sessions (every ~20 samples) volumetrically by weight (Fisher Science scale S94793A; USA) before and after running one of the samples (volume and time monitored). Flow rates between ~20 and 150 μL per min were used for dense and sparse host counts and characterization while flow rates of ~10 μL per min were used when counting viruses. Samples were prepared and stored in the dark at ~18 °C during all flow cytometry. Instrument settings were standardized by running Spherotech Ultra Rainbow Fluorescent Particles (3.0–3.4 μm) beads (Spherotech URFP01-30) before and after each session to ensure that all counts were directly comparable. All gates were applied and counts conducted using FlowJo 7.6 (https://www.flowjo.com; Supplementary Fig. 12).

***E. huxleyi* cell abundance.** *E. huxleyi* were quantified by flow cytometry using chlorophyll autofluorescence between ~10^1 and 10^7 cells per milliliter (Supplementary Fig. 13a). *E. huxleyi* were gated a priori using healthy lab cultures grown in *f*/2 media gated as log_10-transformed 488/692 ± 40 nm excitation/emission plotted against log_10-transformed Perpendicular Forward Scatter (Supplementary Fig. 12). Gates were set to optimize capturing healthy *E. huxleyi* cells while avoiding doublets and false-positive counts from debris in collapsed populations. Finally, gates were tailored in seawater incubations to capture stressed *E. huxleyi* cells when chlorophyll autofluorescence dropped orders of magnitude after weeks at high host densities. In Experiments I and III, *E. huxleyi* were counted in samples fixed with glutaraldehyde (0.5% final concentration) at ~18 °C for 30 min and then flash frozen in liquid nitrogen and kept frozen until analysis. In all other experiments, *E. huxleyi* counts were conducted on fresh, unfixed samples. Counts were then normalized to cell densities (hosts per milliliter) using flow rates calculated circa hourly (above; Fig. 1a).

**Determination of dead cells.** Dead cells were enumerated by flow cytometry in Experiments IV, V, VI, and VII using the live-dead stain SYTOX Green (Thermo Fisher S7020, USA; "Sytox"). In this assay, SYTOX (stock concentration: 5 mM) was added to each sample at a final concentration of 1 μM, and incubated in the dark for ~30 ± 10 min prior to flow cytometric analysis[28]. Events in the *E. huxleyi* chlorophyll autofluorescence gate (see above) were gated through to these SYTOX gates such that only *E. huxleyi* cells were analyzed as SYTOX-positive or -negative, meaning that SYTOX-positive cells were recently dead (Supplementary Fig. 12). Routinely used FITC gates were applied to log_10-transformed 488/520 ± 15 nm excitation/emission fluorescence plotted against log_10-transformed Perpendicular Forward Scatter plots in FlowJo to exclude almost all cells in healthy cultures (<1% false positives were allowed). SYTOX-positive cells in a sample were then

calculated as a percentage of the total *E. huxleyi* population. Host densities below 10^3 cells per milliliter yielded noisy and false-positive percent dead rates (Supplementary Fig. 13b), so only values with host densities greater than this threshold were analyzed.

**Detection of autophagy.** In Experiments IV and V, we probed the lysosomal profile signals of cells, a sensitive indicator of cellular stress and infection, by flow cytometry using the lysosomal stain Lysotracker Deep Red (Thermo Fisher L12492, USA; "Autophagy"). Lysotracker stain (stock concentration: 1 mM) was added to each sample at a final concentration of 110 nM, and incubated in the dark for 30 ± 10 min prior to flow cytometric analysis. Events in the *E. huxleyi* chlorophyll autofluorescence gate (see above) were gated through to the autophagy gates such that only *E. huxleyi* cells were analyzed as autophagy-positive or -negative (Supplementary Fig. 12). Gates were applied to log_10-transformed 640/670 ± 30 nm excitation/emission fluorescence plotted against log_10-transformed Perpendicular Forward Scatter plots in FlowJo to exclude almost all cells in healthy cultures (<1% false positives were allowed). The percentage of autophagy-positive cells was calculated from the total *E. huxleyi* population. Host densities below 10^3 cells per milliliter were shown (Supplementary Fig. 13c) to yield noisy and false-positive percent autophagy-positive rates, so only values with host densities greater than this threshold were analyzed.

**UV-induced autofluorescence.** In Experiments IV and VI, we observed increased UV-excited autofluorescence signatures of cells undergoing lytic infection (355 nm/460 ± 50 nm excitation/emission) by flow cytometry. This signature was observed prior to viral-mediated host collapse and concomitant with viral production in ~500 samples; hence, UV autofluorescence is an inherent cellular characteristic and passive marker of infection. Events in the *E. huxleyi* chlorophyll autofluorescence gate (see above) were gated through to the UV autofluorescence gates such that only *E. huxleyi* cells were analyzed as UV autofluorescence-positive or -negative (Supplementary Fig. 12). Gates were applied to log_10-transformed 355/520 ± 15 nm excitation/emission fluorescence plotted against log_10-transformed Perpendicular Forward Scatter plots in FlowJo to exclude almost all cells in healthy cultures (<1% false positives were allowed). The percentage of UV autofluorescence-positive cells was calculated from the total *E. huxleyi* population (Fig. 2b). Host densities thresholds similar to Sytox and Lysotracker were applied.

**Photochemical quantum yield.** We quantified the photochemical quantum yield of photosystem II ($F_v/F_m$), a sensitive diagnostic marker of photosynthetic stress. $F_v/F_m$ was measured in Experiment IV and VI cultures after ~10 min in darkness using a custom-built mini-FIRe fluorometer[65] with 100 msec sample delay, 20 independent replicate measures per sample, maximum PAR (μmol photons m^−2 s^−1) of 500 with 10 PAR steps; gains were automatically or manually changed to accommodate different sample densities and filter set accurate to host densities of ≥10^3 cells per milliliter. Host density thresholds similar to SYTOX and Lysotracker were therefore applied. Dense cultures were diluted with filtered, autoclaved seawater to ensure accurate readings.

**Extracellular virus abundance.** The concentration of extracellular viruses was determined by quantitative PCR (qPCR). Samples were fixed with betaine (~7% final concentration) at ~18 °C for 30 min and then flash frozen in liquid nitrogen and kept frozen until analysis. To isolate extracellular viruses, 100 μL subsamples were taken from vortexed betaine-fixed samples that had been defrosted at room temperature and centrifuged in PCR strips (15 min; 10,500 x *g*; 4 °C; F45-48-PCR rotor in an Eppendorf 5417R centrifuge; Eppendorf, Germany). Then, 50 μL of host-free supernatant was removed and transferred to 96-well PCR plates (Fisherbrand plates 14230232, sealed with Bio-Rad Microseal "B" seals MSB1001). Any free DNA was removed by DNAse I treatment (1 U per 50 μL reaction; 20 U per milliliter) at 37 °C for 30 min. Before and after incubations, all samples and enzymes were gently centrifuged in a salad spinner (OXO, USA; Item 1155901). All molecular biology incubations were conducted in a TGradient Thermocycler (Biometra, Germany). Viruses were lysed by incubation at 95 °C for 60 min ("boil prep" and DNAse denaturation) followed by three freeze/thaw cycles of 0 °C for 10 min to 95 °C for 10 min. Plates were then unsealed, Proteinase K was added at a 0.2 μg μL^−1 final concentration, and incubated at 37 °C for 60 min. Proteinase K was then denatured at 95 °C for 20 min after which samples were diluted tenfold with molecular grade water (Invitrogen nuclease-free water; AM9930) and stored at −20 °C. Viral copies were quantified by qPCR targeting the viral major capsid protein (MCP) using a Mx3000P qPCR thermocycler (Stratagene, USA) with 10 μL reactions with qPCR plates and caps (Applied Biosystems MicroAmp Optical plates N8010560 with Fisherbrand caps 14230230). Reactions (10 μL total volume) were composed of 5 μL of Power Up SYBR Mix (Applied Biosystems Power Up SYBR Green Master Mix; A25741), 0.3 μL of dimethyl sulfoxide, 0.125 μL (125 nM final concentration) of forward and 0.25 μL (250 nM final concentration) of reverse primer working stocks[35] (primers prepared by diluting Integrated DNA Technologies (USA) primers to 100 μM with ~800 μL of molecular water, and diluting again tenfold to 10 μM working stocks), 3.4 μL of molecular water and 1 μL of template. MCP forward primer: 5′-TTC GCG CTC GAG TCG ATC-3′; MCP reverse primer: 5′-GAC CTT TAG GCC AGG GAG-3′[35]. Primer concentrations

were optimized by amplifying known template (~100 copies per reaction) with a matrix of forward and reverse primers from 125 nM to 1 μM and annealing temperate optimized by gradient PCR. Reactions were run at 95 °C for 10 min, then 40 cycles of 53 °C for 1 min, 72 °C for 1 min, 95 °C for 1 min. Products were confirmed by dissociation curves. Amplification curves and cycles to threshold ($C_t$ values) were translated to copies per reaction (and therefore viruses per milliliter of original samples) using an internal standard curve run in each qPCR plate. Standard curves were generated from lysates (~$10^8$ viruses per milliliter) processed similar to samples (DNAse, Proteinase K, boil prep, etc.), diluted 100-fold (more concentrated lysates showed qPCR inhibition), and then serial diluted by eight times by half. This gave a standard curve of $3 \cdot 10^3$, $1.5 \cdot 10^3$, $7.5 \cdot 10^2$, $3.75 \cdot 10^2$, $1.87 \cdot 10^2$, 94, 47, 24, and 12 copies per reaction (Supplementary Fig. 13d). Samples below the range of the standard curve with ≤10 copies per reaction were discarded, effectively giving a detection limit of $10^5$ viruses per milliliter in the original samples. Each plate also had three wells dedicated to no-template controls to detect contamination and non-specific amplification.

**Frequency of infected cells**. A dilution approach was used to estimate the fraction of cells infected in 10:1 virus:host MOI coincubation and pre-infected treatments (Experiment V; Supplementary Fig. 4). Five hours after first mixing viruses and hosts and ~3 h after cultures were diluted to experimental densities and after the onset of the lytic program[66], cells were diluted 1000-fold in new $f/2$ media to preclude subsequent rounds of new infection, and counted after 24 h. While dilution of cultures within ~2 h of viral addition gives the temperate phenotype described in Fig. 1, diluting infected cells after the initiation of the lytic program but before death, occuring at ~5 h after mixing, allows the quantification of infection without encountering subsequent infections consistent with a one-step infection curve. Comparing host densities before and after incubation yielded an estimate of infected cells, which was normalized as a percent of the total host population.

**Satellite estimation of *E. huxleyi* densities**. The global distribution of *E. huxleyi* cell abundance (cells per milliliter) was estimated using satellite ocean color fields of particulate inorganic carbon (PIC)[67,68], together with empirical relationships between PIC, coccolith abundance, and *E. huxleyi* density. In order to determine the upper bound of observed cell densities, the maximum observed PIC at each pixel during the period 2003-2017 was determined from global Level 3 (~9 km) 8-day composite MODIS-Aqua PIC. Retrieved PIC concentration (milligram per liter) was first converted to an equivalent coccoliths per milliliter, which were subsequently associated with *E. huxleyi* densities. Specifically, Balch et al.[69] report values of all three quantities and their inter-relationships measured during a large *E. huxleyi* bloom in the summertime North Atlantic Ocean. Application of these relationships to our data convey some practical constraints, like a PIC limit below which estimated cell densities become negative, but should not affect our estimate of potential maximum cell concentration.

**Field mesocosm experiments**. Mesocosm experiments were conducted at the University of Bergen Marine Biological Station in Espegrend, Norway from 3 to 20 June 2008. To stimulate a bloom and drive up *E. huxleyi* cell densities, mesocosms were enriched with nutrients in Redfield ratio stoichiometry, involving daily additions to triplicate enclosures at 1.5 μM NO$_3$: 0.1 μM PO$_4$; N:P = 15[31]. *E. huxleyi* counts were conducted immediately on site in Espegrend, Norway using methods similar to those above for laboratory studies. Cells were counted on a FACScan flow cytometer (Becton Dickinson, USA) equipped with a 15 mW laser exciting at 488 nm and with a standard filter set up[54]. Samples were analyzed at a high flow rate (~70 μL per min) and specific phytoplankton groups were discriminated by differences in their forward or right angle light scatter. Virus-like particles (c.f., EhVs-only quantified with qPCR above) were enumerated at the Rutgers University Microbial Flow Sort Facility (https://marine.rutgers.edu/microbial-flow-sort-lab/) using a BD Influx Mariner 209s. Briefly, glutaraldehyde-fixed samples (0.5% final concentration) that had been flash frozen after 30 min fixation time were thawed, diluted 50-fold with 0.2 μm filtered 1 part SYBR Gold per 20,000 parts Tris-EDTA, incubated at 80 °C for 10 min, cooled, vortexed, and run on the flow cytometer with routinely used log$_{10}$-transformed 488 nm/542 ± 27 nm excitation/emission plotted against log$_{10}$-transformed Perpendicular Side Scatter gates[70]. Intracellular viral transcription activity (relative transcription) was extracted from Fig. 2 in Pagarete et al.[55] using Web-PlotDigitizer (https://automeris.io/WebPlotDigitizer/).

**Extracting empirical parameters**. Empirical and modeled parameters were compared for "fit" of virulent and temperate models using the density at which viruses initiated lysis (lytic density; Fig. 4b). Lytic densities were defined as the highest observed density in each infected culture that showed viral-mediated declines or where they first diverged from uninfected controls, and was assessed on a flask-by-flask basis (Supplementary Fig. 7). Note that actual maxima may have been missed between sampling periods in lab data points, but not in theoretical prediction points. Carrying capacity was calculated in $f/2$ systems as the maximum density observed in uninfected cultures that reached stationary phase ($6.60 \cdot 10^6 \pm 3.15 \cdot 10^5$ cells per milliliter). Carrying capacity was estimated in uninfected seawater cultures in each density and in each experiment independently. Further, seawater

showed less well-defined carrying capacities compared to $f/2$ incubations (Supplementary Fig. 5).

**Statistical analyses and graphs**. Graphs were plotted using the ggplot2 package in R version 3.4.1 "Single Candle" (https://www.r-project.org/). Locally estimated scatterplot smoothing (LOESS) lines were applied with the ggplot2 stat_smooth function. Plots made in R were combined and finished in Inkscape (https://inkscape.org/).

**Quantiative comparison between data and models**. The models' match to the data were evaluated using the AIC[46] and using the absolute magnitudes of the residuals to calculate the difference between the model predictions and data values. This was done for experiments conducted with $f/2$ rich media by pooling all data sets generated with this media and comparing to the models. Seawater experiments had different carrying capacities in each experiment, precluding pooling growth curves from different experiments. As a result, we did not apply the AIC analysis to these data sets due to a lack of replication and power without pooling. The AIC was here defined as $\text{AIC} = 2M + N \log(\text{RSS}/N)$, where $M$ is the number of parameters for a given model, RSS is the residual sum of squares, and $N$ is the sample size. We consider residuals of the log-transformed data because the data are logarithmically distributed and because both growth and death are multiplicative processes. The use of the second term on the right hand side of the above equation is in analogy with the case of normally distributed residuals, though we note here that the residuals are in almost all cases not normally distributed according to standard statistical tests (e.g., ref. [47]). We chose to use the expression, however, in lieu of a better option; this choice does not affect our general message, because the differences between the virulent model and observations are so quantitatively and qualitatively large for most cases. The AIC differences in Fig. 3 show the relative performance of virulent and the phenomenological temperate model, supporting that temperateness explains the behavior observed for *sparse* host densities, while virulent and temperate dynamics seem mostly indistinguishable for *crowded* densities.

Nonetheless, due to the non-normality of residuals as an additional test, we further evaluated the closeness of each model to our experimental results by measuring the MAE, the sum of the absolute values of the (log-transformed) residuals divided by $N$. This statistic penalizes residuals very differently from the AIC, because it is proportional to their amplitude rather than the square of their amplitude. As Fig. 3 shows, the MAEs confirm our conclusion above. See Supplementary Notes 1 and 2 for further details.

**Virulent and temperate dynamical models**. To better understand the mechanisms underlying the host–virus dynamics observed in our experiments, we compared three versions of a host–virus interaction model (Supplementary Figs. 2 and 5): (i) a classic version in which the virus is purely virulent; (ii) a temperate version of the classic model in which induction occurs at times informed by the host physiological data from our experiments; (iii) an improved version of the temperate model that replaces the pre-set induction times by times that emerge from suggested mechanisms for the self-regulation of induction. The comparison between (i) and (ii) aim to discern whether the behavior observed in the laboratory can be explained with a purely virulent virus or with a temperate one. Further, the introduction of (iii) aims to shed some light onto the mechanisms underlying induction. We summarize here the first two; see Supplementary Table 2 and Supplementary Figs. 2 and 5 for further details (model parametrization), and Supplementary Notes 1 and 2 for the third version of the model.

Equations (1–3) in Fig. 6 represent model versions (i) and (ii), with the growth of the uninfected population presented in black, the additions for the classic virulent model in red, and the additions to represent temperate dynamics in blue, describing the dynamics of uninfected host [H], free infective virus [V], and

$$\frac{d[H](t)}{dt} = \mu(t)[H] - m[H] - k[H][V] + \mu_I(t)[I] \qquad (1)$$

$$\frac{d[V](t)}{dt} = r_s B k_L[I] - k([H] + [I])[V] - m_V[V] \qquad (2)$$

$$\frac{d[I](t)}{dt} = k[H][V] - r_s k_L[I] - m[I] \qquad (3)$$

**Fig. 6 Dynamic equations describing all versions of our models.** [H], [V], and [I] represent densities of uninfected hosts, viruses, and infected hosts, respectively. The growth of a host population in the absence of viruses is presented in black; the additions for the classic virulent model in red; and the additions to represent temperate dynamics are presented in blue. See text for further description of terms and parameters.

infected host [I] concentrations (all in units of individuals per liter, see Supplementary Table 2). In the first equation (dynamics of the uninfected host population), the first term represents population growth; the second term represents natural mortality; the third term represents infection events, which occur at a rate $k$ (viral adsorption rate); the last term assumes that infected hosts can reproduce similarly to uninfected cells if the infecting virus is temperate (which, for simplicity, we assume results in new uninfected hosts). In the second equation (dynamics of the extracellular viral population), the third term represents viral decay in the extracellular milieu; the second term represents infection events (including the possibility of superinfection wherein several viruses attaching to or passively infecting the same host); and the first term represents the viral offspring resulting from lysis (which only occurs if the virus is virulent or is a temperate virus undergoing induction, see below); each virus produces $B$ virions per host, and we assume here that the offspring is released at a lytic rate $k_L$ (inverse of the infection time; latent period, $L$). In the last equation (dynamics of infected hosts), the first term represents infection events; the second term represents lysis of hosts by virulent viruses (or temperate viruses undergoing induction); and the third term represents host natural mortality.

Based on the experimental data for the uninfected treatment, the models use a phenomenological logistic equation to implement host growth rate. Specifically, the following expression provides a good approximation to the uninfected population growth rate:

$$\mu(t) = \mu_{\text{eff}}(t)\left(1 - \frac{[H](t) + (1 - r_s)[I](t)}{K}\right) \quad (4)$$

where $K$ represents the carrying capacity (see Supplementary Table 2 for parameter values and units), and:

$$\mu_{\text{eff}}(t) = \begin{cases} \mu_{\max} & \text{if} \quad t < 2 \\ s_\mu t + n_\mu & \text{if} \quad 2 < t < t_\mu \\ \mu_{\min} & \text{if} \quad t > t_\mu \end{cases} \quad (5)$$

that is, the growth rate stays at a maximum level for two days, then decreases linearly to reach a minimum level at $t_\mu$. See Supplementary Table 2 for more details, including the values for $s_\mu$, $n_\mu$, and $t_\mu$. Equation (4) aims to replicate as closely as possible the growth conditions for the uninfected host population, including changes in its host growth rate due to unknown/uncharacterized sources of physiological stress for which we may have no information. Note that the term takes into account that the "healthy-like" behavior of infected cells before induction reduces the available nutrient for the total host population. However, our results do not change qualitatively if, for example, we replace Eq. (4) with a standard Monod growth function dependent on, e.g., nitrogen as single source of growth limitation in our simulated batch experiment. We further assume that hosts infected with temperate viruses continue their usual life cycle and, therefore, replicate. Thus, the growth rate of infected hosts is:

$$\mu_I(t) = (1 - r_s)\mu(t) \quad (6)$$

Given $r_s$ below (Eq. (7)), infected hosts replicate at the same growth rate as healthy hosts while the virus is temperate, and do not replicate at all when the virus is virulent or undergoing induction as the virus utilizes the synthesis machinery of the host, precluding host replication. We assumed for simplicity that infected host replication produces only uninfected hosts. See Supplementary Notes 1 and 2 for further details and discussion on other options.

Induction, wherein temperate viruses enter lytic replication, is implemented in the equations above via a switch function:

$$r_s = \begin{cases} 1 & \text{if virus is virulent} \\ 0 & \text{if virus is temperate} \end{cases} \quad (7)$$

Following our experimental data, we assumed that the default mode of the virus is temperate, with a physiologically dependent induction switch that we modeled in two different ways. Here, we discuss the phenomenological implementation (model (ii)) whose results are shown in Fig. 3 and Supplementary Fig. 8, but see Supplementary Notes 1 and 2 for the more complete version of the model (model (iii)).

For the phenomenological temperate model, we used the decline in the photochemical quantum yield curve as a quantitative indicator of stress (see $F_v/F_m$ curve, Fig. 2c), by imposing an induction time, $t_s$, matching the beginning of the decline in that curve. Thus, in this version of the model, $r_s = 1$ for $t \geq t_s$, and zero otherwise. This implicitly assumes that the virus does not switch back to the temperate mode in the duration of the experiment, which is consistent with the initial increase and decline of the host population. As shown in Fig. 3 and Supplementary Fig. 5, the resulting behavior obtained with this simple temperate version is qualitatively and quantitatively similar to that from the experiments. As in experiments, the specific time $t_s$ is assumed to depend on initial host density, but not considerably on the treatment (see Supplementary Notes 1 and 2). The virulent, purely lytic mode can be seen as a particular implementation of this case in which induction occurs from the outset, where $t_s = 0$.

As explained in Supplementary Notes 1 and 2, before we introduced a temperate mode, we tested whether the delay in the release of the virus could be explained by explicitly including the latent period in our virulence model. To that end, we introduced a delayed version of the classic virulent model[71]. The resulting

curves show a qualitative behavior that is similar to that of the classic virulent virus, including the fact that the host population directly declines for the pre-infected treatment that includes additional viruses from the outset. Only decreasing the probability of successful contacts/infections allowed the host population to grow in this simulated treatment, but with a week-long delay that contrasts with the healthy-like growth of the experimental population observed in this case. Thus, a delayed virulent model cannot explain the behavior observed in the laboratory. We also explored the possibility for host physiology to affect the viral latent period and/or burst size instead[72], which did not qualitatively alter the behavior of the virulent model. In addition, we considered exclusive infection (as opposed to superinfection) where an infected host cannot be infected by more than one virus. This variation, which can be implemented by replacing the second term in Eq. (2) (Fig. 6) with $k[H][V]$, meaning that free viruses attach and infect only uninfected hosts, did not qualitatively alter our results. Finally, although the parametrization used here is a very conservative representation of EhV trait values (see Supplementary Table 2), other parametrizations (e.g., lower contact rates or longer latent periods) did not qualitatively alter our conclusions, as summarized in Supplementary Table 3.

**Spatially explicit encounter rates calculation**. We estimated the time for a virus to find a host using the model of the encounter between two particle types[73],

$$E = \beta C_V C_H \quad (8)$$

where $E$ is the volumetric encounter rate, $\beta$ is the encounter rate kernel, $C_V$ is the concentration of virus, and $C_H$ is the concentration of host (note the change in notation with respect to the models, to emphasize the fact that these are expected values for concentrations measured in the field). Rearranging Eq. (8) provides the time for a virus to encounter a host cell,

$$\frac{C_V}{E} = \frac{1}{\beta C_H} \quad (9)$$

In the dynamic models for the laboratory observations, we assumed that the encounter rate kernel was simply a constant, $k$, which is a good approximation for our laboratory setup. The more general form of the encounter kernel $\beta$, however, differentiates between host–virus encounters due to Brownian motion $\beta_b$, differential sinking $\beta_s$, and turbulence $\beta_t$,

$$\beta = \beta_b + \beta_s + \beta_t \quad (10)$$

The encounters due to Brownian motion depend on the sizes of virus and host,

$$\beta_b = \frac{2}{3}\frac{k_B T}{\eta}\frac{(r_V + r_H)^2}{r_V r_H} \quad (11)$$

where $k_B$ is the Boltzman's constant, $T$ is absolute temperature, $\eta$ is dynamic viscosity, $r_V$ is viral radius, and $r_H$ is host cell radius. The encounters due to differential sinking are estimated assuming that viral sinking is negligible,

$$\beta_s = \pi w_H (r_V + r_H)^2 \quad (12)$$

where $w_H$ is the terminal sinking velocity of the host cell. We calculated $w_H$ using Stokes' Law for small spheres, assuming that the host cells have a diameter of 5 and 6 μm for naked and calcified cells, respectively, and densities of 1.05 and 1.19 g cm$^{-3}$ for naked and calcified cells[74], respectively. Finally, the encounters due to turbulence are

$$\beta_t = 1.3\left(\frac{\varepsilon}{\nu}\right)^{1/2}(r_V + r_H)^3 \quad (13)$$

where $\varepsilon$ is the dissipation rate of turbulent kinetic energy and $\nu$ is the kinematic viscosity.

We estimated the time for a virus to encounter a host in relatively calm water ($\varepsilon = 10^{-8}$ m$^2$ s$^{-3}$) and in strong turbulence ($\varepsilon = 10^{-4}$ m s$^{-3}$), equivalent to near-surface conditions under moderate winds. Depending on the host cell's calcification state and on dissipation rate, encounter rates are dominated by host cell sinking ($\beta_s$) for calcified cells or by turbulence ($\beta_t$) for naked cells (see Supplementary Fig. 10b, c).

Encounters between particles are central to many ecological processes, but the $\beta$ kernels take diverse functional forms. Additive $\beta$ kernels (Eq. (9)) are widely used to estimate encounter rates among planktonic predators and prey[75] and in coagulation models[73]. In viral ecology, however, encounter rates are more often estimated using an advection-diffusion framework based on the Sherwood number $Sh$[76], a ratio of the contact rates due to advection and diffusion to the contact rates due to diffusion alone. The $Sh$ framework is particularly appropriate when both particle types are small (~1 μm) and encounter rates are strongly influenced by Brownian motion. For the $E.\ huxleyi$-EhV system in nature, however, the host cells sink at speeds that render Brownian motion a negligible contributor to encounters. Moreover, there is no straightforward modification for encounters in turbulence, which is a fundamental condition underlying the infection process. In general, encounter rates are slightly higher when estimated from additive kernels than from $Sh$-based kernels, and, thus, our use of Eq. (10) provides conservative estimates of the time for a virus to encounter a host (Eq. (9)).

**Reporting summary**. Further information on research design is available in the Nature Research Reporting Summary linked to this article.

## Data availability

All data sets and mathematical models are available at github ([https://github.com/benjaminwilliamknowles/Coup-de-Grace](https://github.com/benjaminwilliamknowles/Coup-de-Grace)). Source data are provided with this paper.

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

## Acknowledgements

This paper is dedicated to the memory of Merry Youle, viral ambassador. This study was supported by grants to KDB from the National Science Foundation (NSF; OCE-1061883, OCE-1537951, OCE-1559179 [to K.D.B. and K.T.]), NASA (15-RRNES15-0011, 80NSSC18K1563), and the Gordon and Betty Moore Foundation (Award # 3789), as well as postdoctoral fellowship to B.K. from the Institute of Earth, Ocean, and Atmospheric Sciences at Rutgers University. M.J.B., K.H., J.R.G., and T.K.W. were supported by the NASA North Atlantic Aerosol and Marine Ecosystem Study (NAAMES; Award #NNX15AF30G) and K.H. by NAAMES NASA Award #NNX15AE70G. C.A.C. was supported by NSF Award OCE-1537943. S.V. was supported by TMS Starting grant TMS2018REC02. We acknowledge the captain and crew of the *R/V Knorr*, the Marine Facilities, and Operations at the Woods Hole Oceanographic Institution, the support staff at the University of Bergen's Marine Biological Station (Espegrend, Norway), and members of the NA-VICE and 2008 Bergen mesocosm science teams for their role in collecting and processing field samples and data used in this study. We would like to thank Maeve Eason-Hubbard, Steve Giovanonni, Austin Grubb, Elizabeth Harvey, Chana Kranzler, Christien Laber, and Jason Latham for insightful conversations and ideas that helped guide this project.

## Author contributions

B.K. and K.B. designed the project. B.K., J.A.B., N.C., B.D., H.F., L.H., C.T.J., F.N., J.I.N., B.S., and T.W. conducted experimental and theoretical/computational work. All authors (B.K., J.A.B., M.J.B., K.G.B., B.B.C., C.A.C., N.C., B.D., H.L.F., J.R.G., J.A.G., K.H., L.H., C.T.J., F.N., J.I.N., B.S., K.T., T.F.T., S.V., C.W., T.K.W., K.D.B.) helped develop ideas and provided critical insight on the manuscript.

## Competing interests

The authors declare no competing interests.
