## [Peer Review File · Nature Communications]

REVIEWERS' COMMENTS:

Reviewer #2 (Remarks to the Author):

The authors made a big effort to answer the comments raised by the reviewers. Yet, the manuscript remains relatively difficult to follow with a multitude of different experiments. I am not sure what should be done to improve that at this stage. The main problem stems from the fact that the authors have only access to measures at the population level. The demonstration of temperate infections at the cellular level would be very helpful. For instance, to show that the virus is in the cell but somehow, it does not trigger a lytic cycle before a certain signal. But I realise how difficult this single-cell analysis would probably be.

Also, regarding the adaptive nature of induced lysis, you may want to refer to previous analysis that have shown when this type of environmentally triggered induction is adaptive (e.g.

<https://pubmed.ncbi.nlm.nih.gov/25244050>

<https://pubmed.ncbi.nlm.nih.gov/26946976/>).

1 REVIEWERS' COMMENTS:

2 Reviewer #2 (Remarks to the Author):

3 The authors made a big effort to answer the comments raised by the reviewers.

4 We thank the reviewer for their comments on previous versions of the manuscript.
5 Incorporating their comments, as well as those of the other reviewers, allowed us to
6 significantly improve the manuscript, particularly its interest to a broad audience.

7 Yet, the manuscript remains relatively difficult to follow with a multitude of different
8 experiments. I am not sure what should be done to improve that at this stage.

9 Identifying the hitherto overlooked dynamic of temperate infection in this system, and
10 coming to understand how this oversight occurred, required us to assemble many
11 independent lines of evidence. We have sought to make the narrative and examination
12 of evidence as linear and straightforward as possible to ensure its interest across a
13 broad audience. We have also provided a schematic figure that outlines the logical
14 steps in our experiments to make sure readers can follow our thought process and
15 inference (Supplementary Figure 3) in addition to summaries of the treatments in each
16 of our independent experiments (Supplementary Table 1).

17 The main problem stems from the fact that the authors have only access to measures at the
18 population level.

19 The flow cytometry that forms the 'backbone' of our work provides cell-by-cell measures
20 of infection state in our experiments. As a result, the majority of our manuscript is
21 fundamentally underpinned by individual cell-level analyses using an array of
22 diagnostic/physiological staining. Indeed, the reviewer highlights one of the main
23 strengths of our work; we were only able to diagnose temperate infection precisely
24 because we conducted cell-by-cell analyses.

25 The demonstration of temperate infections at the cellular level would be very helpful. For
26 instance, to show that the virus is in the cell but somehow, it does not trigger a lytic cycle before
27 a certain signal. But I realize how difficult this single-cell analysis would probably be.

28 The demonstration of temperate infection at the cellular level, where lysis is initiated
29 only after induction is triggered by a cellular stress signal, is a major strength of our
30 paper. We provide independent lines of evidence that infection – but not lysis – has
31 taken place in our low initial density cultures until they reach high densities. These
32 include:

- 33 1) The rapid death of cells in high density cultures but not in low density in our
34 'Preinfected' treatment, where infection was conducted at high host densities and
35 extra-cellular viruses removed prior to dilution to experimental initial densities. If
36 the viruses weren't inside the cells prior to dilution, then death would not be
37 observed in any cultures (**Figure 1**).
- 38 2) Death observed in cells diluted to low densities after the onset of the lytic
39 program, that showed that ~ 50 % of cells or more were infected in our

40 'Preinfected' treatment (**Supplementary Figure 4**), where lysis was only
41 observed in high density cultures, and only took place in initially low density
42 cultures after they had reached high densities when the viruses 'chose' to initiate
43 lysis (**Figure 1**) with the onset of stress (**Figure 2**). This was also shown by
44 theoretical modeling based on a stress induction trigger (**Figure 3**).

- 45 3) Viruses initiated lysis differentially in response to cellular physiology, as shown
46 by our rich media vs. seawater experiments. This shows further the impact of
47 physiology on induction of lysis by the viruses (**Figure 4**).
- 48 4) Finally, it is important to note that – until the onset of lysis – infected cells were
49 indistinguishable from their uninfected controls either by growth dynamics
50 (**Figure 1**) or by cellular (cell-by-cell measured) stress markers (**Figure 2**). This
51 single-cell resolution monitoring of dormant infection is also consistent with
52 temperate infection.

53 Taken together, our experiments provide robust, reproducible evidence of viral infection
54 leading to lysis only when a physiological trigger (cellular stress, as shown by multiple
55 cellular-level flow cytometry data sets) signal is observed.

56 Also, regarding the adaptive nature of induced lysis, you may want to refer to previous analysis
57 that have shown when this type of environmentally triggered induction is adaptive (e.g.
58 <https://pubmed.ncbi.nlm.nih.gov/25244050>
59 <https://pubmed.ncbi.nlm.nih.gov/26946976/>).

60 We'd like to thank the reviewer for suggesting these very interesting references. We
61 have now incorporated them and their points in the Introduction (lines 64 – 67 and 157-
62 158).